# ComboGAN: Unrestrained Scalability for Image Domain Translation

**Asha Anoosheh**
Computer Vision Lab
ETH Zürich
ashaa@ethz.ch

**Eirikur Agustsson**
Computer Vision Lab
ETH Zürich
aeirikur@ethz.ch

**Radu Timofte**
ETH Zürich
Merantix GmbH
timofter@ethz.ch

## Abstract

This past year alone has seen unprecedented leaps in the area of learning-based image translation, namely the unsupervised model CycleGAN, by Zhu *et al.*. But experiments so far have been tailored to merely two domains at a time, and scaling them to more would require an quadratic number of models to be trained. With two-domain models taking days to train on current hardware, the number of domains quickly becomes limited by training. In this paper, we propose a multi-component image translation model and training scheme which scales linearly - both in resource consumption and time required - with the number of domains.

## 1 Introduction

Introduced by Zhu *et al.*, CycleGAN (Zhu et al. (2017)) uses a conditional setting where Generative Adversarial Networks (GANs) (Goodfellow et al. (2014)) create a framework for unsupervised image-to-image translation, meaning no alignment of image pairs are necessary. CycleGAN consists of two pairs of neural networks, $(G, D_A)$ and $(F, D_B)$, where the translators between domains $A$ and $B$ are $G : A \to B$ and $F : B \to A$. $D_A$ is trained to discriminate between real images $a$ and translated images $F(a)$, while $D_B$ is trained to discriminate between images $b$ and $G(a)$. The system is trained using both an adversarial loss and a cycle consistency loss, the latter regularizing the unconstrained problem of translating in a single direction, by encouraging the mappings $G$ and $F$ to be inverses of each other such that $F(G(a)) \approx a$ and $G(F(b)) \approx b$.

The concurrent work of StarGAN (Choi et al. (2017)) aimed to extend CycleGAN to multiple domains, by having only one generator and discriminator to be shared by all domains. While this proved satisfactory for translation among a limited number of human-face attributes, this approach is, in theory, unsustainable for large numbers of domains or for domains with large variation.

## 2 The ComboGAN Model

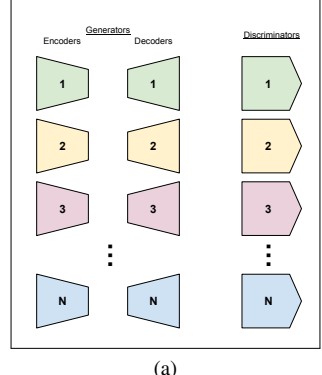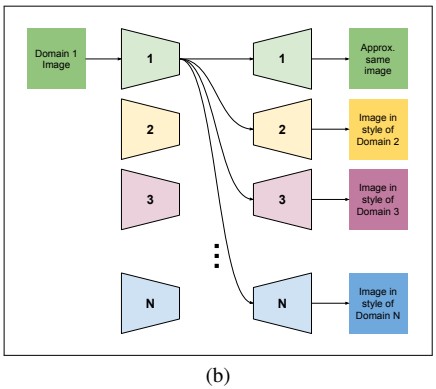

(a)                        (b)

Figure 1: ComboGAN: (a) Model design setup for N domains. (b) Example inference functionality of translation from one domain to all others.

**Decoupling the Generators**   The scalability of setups such as CycleGAN's is hindered by the fact that both networks used are tied jointly to two domains, one from some domain $A$ to $B$ and the other from $B$ to $A$. To add another domain $C$, we would then need to add four new networks, $A$ to $C$, $C$ to $A$, $B$ to $C$, and $C$ to $B$. To solve this issue of exploding model counts, we introduce a new model, ComboGAN, which decouples the domains and networks from each other. ComboGAN's generator networks are identical to the networks used in CycleGAN (see Appendix A for network specifications), yet we divide each one in half, labeling the frontal halves as encoders and the latter halves as decoders. We can now assign an encoder and decoder to each domain. This approach is similar to the one Google took for multi-language machine translation (Johnson et al. (2016)).

As the name ComboGAN suggests, we can combine the encoders and decoders of our trained model like building blocks, taking as input any domain and outputting any other. For example during inference, to transform an image $x$ from an arbitrary domain $X$ to $y$ from domain $Y$, we simply perform $y = G_{YX}(x) = Decoder_Y(Encoder_X(x))$. With only one generator (an encoder-decoder pair) per domain, the number of generators scales exactly linearly with the number of domains. The discriminators remain untouched in our experiment; the number of discriminators already scales linearly when each domain receives its own. Figure 1 displays our full setup.

**Training**   While ComboGAN's discriminators are trained identically to CycleGAN, the generator training scheme must be adapted. Fully utilizing the same generator losses as CycleGAN requires focusing on two domains, as the generator's cyclic and adversarial training are not directly adaptable for more domains. At the beginning of each iteration, we select two domains $X, Y \in \{1..n\}$ from our $n$ domains, uniformly at random. Then maintaining the same notation as CycleGAN's training loss, we set $A := X$ and $B := Y$ and proceed as CycleGAN would for the remainder of the iteration.

Randomly choosing between two domains per iteration means we should eventually cover training between all pairs of domains uniformly. Though of course the training time (number of iterations) required must increase as well. We keep the training linear in the number of domains, since the number of parameters in our model increases linearly with the number of domains, as well. We only desire each domain - not each pair - to be chosen for a training iteration the same number of times as in CycleGAN. We observe that since a domain $X$ is chosen in each iteration with probability $\frac{2}{n}$, during training it is chosen in expectation $\frac{n}{2} \cdot k_n$ times. Requiring equality to the two-domain case $k_2$, we obtain $k_n = \frac{k_2}{2}n$, or $\frac{k_2}{2}$ iterations per domain, which proves satisfactory in practice.

It is easy to see that for the case of two domains, ComboGAN becomes exactly equivalent to CycleGAN. But for more than two domains, the model must be implicitly placing images into a shared latent space - ideally invariant to all domains. This has potential to be exploited for other tasks.

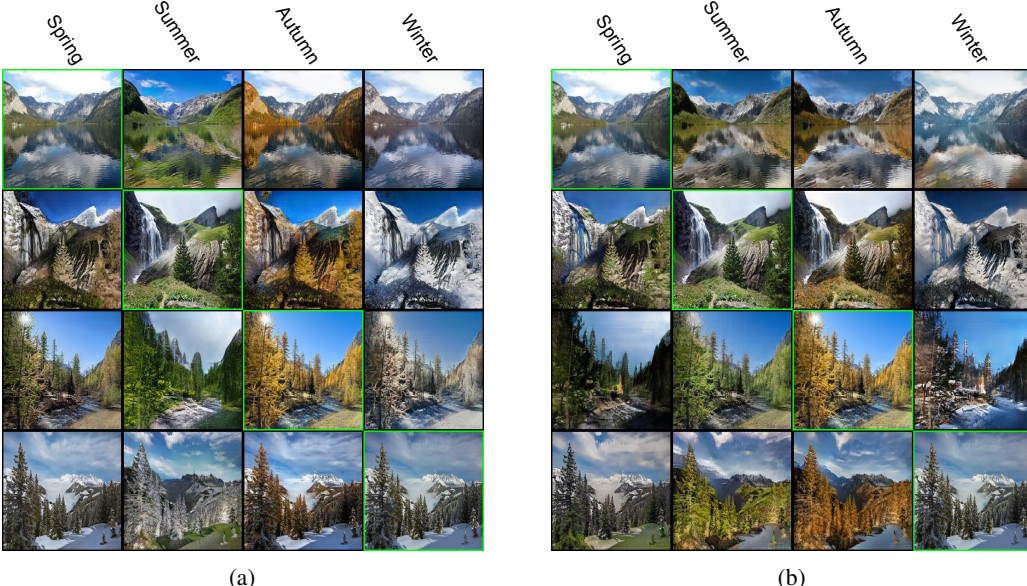

Figure 2: (a) Validation results for pictures of the Alps in all four seasons. Original images lie on the diagonal. (b) Same Alps images but from standard CycleGAN results instead.

# 3 EXPERIMENTAL RESULTS

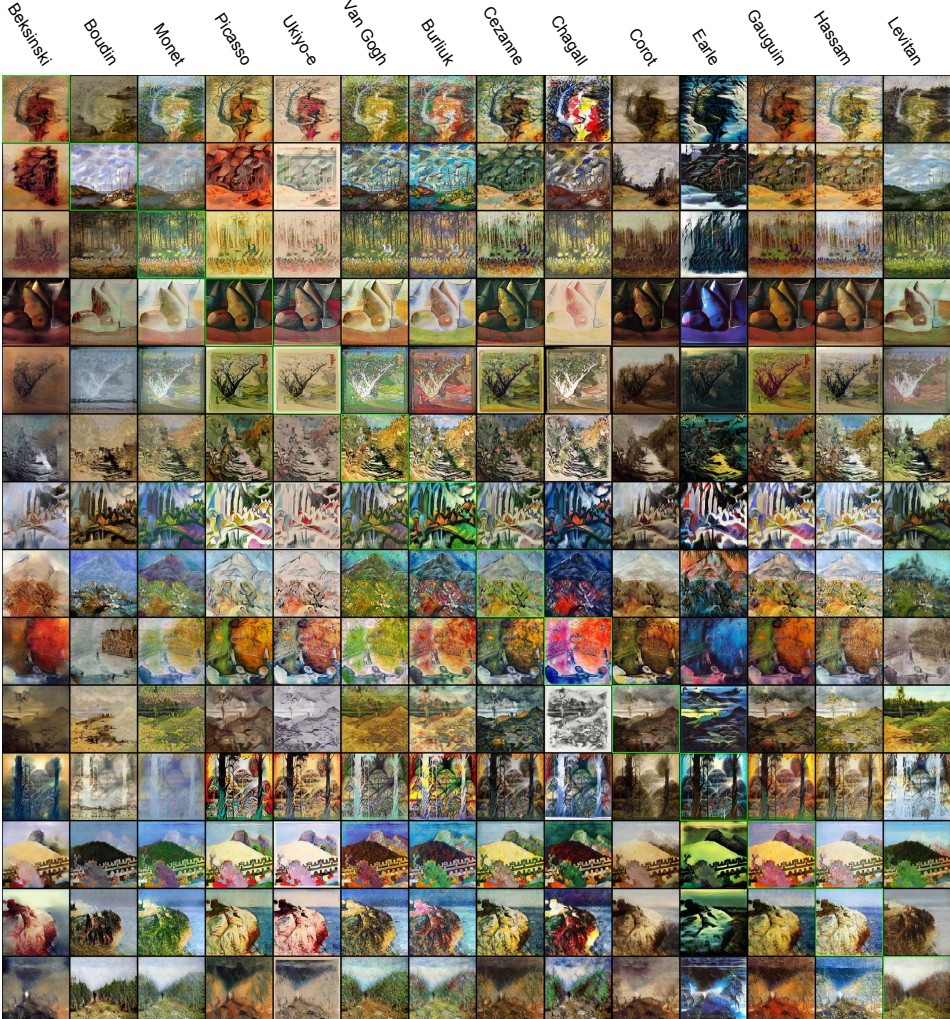

Figure 3: Validation results for our 14 painters. Original images lie on the diagonal.

The first of two datasets used in this experiment consists of approximately 6,000 images of the Alps mountain range scraped from Flickr. The photos are individually categorized into four seasons based on their timestamps. Figure 2 shows validation image results for ComboGAN trained on the four seasons for 400 iterations. The same figure also contains results from CycleGAN trained on all six combinations of the four seasons to produce the same images, demonstrating that Combo-GAN maintains comparable quality, while only training four networks for 400 epochs instead of CyleGAN's twelve nets for a total of 1200 epochs.

The other dataset is a collection of approximately 10,000 paintings total from 14 different artists from Wikiart.org, whose names are listed above Figure 3, which displays randomly-chosen validation images. Looking at columns as a whole, one can see common texture behavior and color palettes common to the pieces per artist column. The fourteen painters dataset ran 1400 epochs in 220 hours on our nVidia Titan X GPU with ComboGAN. Pairwise CycleGAN instead would have taken about 2860 hours, or four months, and required (14 choose 2) = 91 models, thus comparison with CycleGAN is not shown as it is computationally infeasible.

It's noteworthy that the training hyperparameters are unaltered from the original CycleGAN; no modifcations were needed for ComboGAN to train stably every time. Our code is available at https://github.com/AAnoosheh/ComboGAN and our original paper with full details can be found at https://arxiv.org/abs/1712.06909

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
