# OpenReview forum: "ComboGAN: Unrestricted Scalability for Image Domain Translation"
_ICLR.cc/2018/Workshop — Accept_

### Official Review · AnonReviewer2 · 2018-03-05
**A complexity optimization for N to N image style transfer with GANs - lacks objective evaluation**

**Rating:** 6
**Confidence:** 4

**Review:**

This paper proposes to reduce the combinatoric complexity of transferring image styles from N to N domains by decoupling the generators. The approach uses N encoder-decoder pairs instead of N*(N-1). The authors show that the model can be  trained in N*k/2 iterations, where k is the number of iterations required to trained a 2 domains model with CycleGAN. The author show that the generated images are "similar" to images generated by CycleGAN.

Pro:
1) a neat extension of CycleGAN that removes combinatoric explosion for large number of image domains.
Cons:
1) a relatively obvious incremental improvement over CycleGAN
2) it is unclear whether the differences between the images generated by CycleGAN and this are significant or not.

---

### Official Review · AnonReviewer3 · 2018-03-08
**Simple and good**

**Rating:** 7
**Confidence:** 4

**Review:**

This aper propose a method for image translation across multiple domains based on pervious works such as CycleGAN. The main difference being that unlike previous works when the number of models to train increase quadraticly with number of domains, here only increase linearly.

It’s a novel and sensible approach to expand number of domains and the samples look very good. As for significance, the question goes back to significance of image translation methods, which might be more significant for computer vision community than machine learning.

Pros:
Simple to understand
Good results

Cons:
The problem is not well defined for readers that might not be familiar with image translation domain and prior work. But given the limited space for workshop submission is justifiable.

---

### Official Review · AnonReviewer1 · 2018-03-12
**an extension of CycleGAN to multi-domains**

**Rating:** 5
**Confidence:** 3

**Review:**

This paper is a simple extension of CycleGAN to overcome to computational issue when increase the number of domains. It utilizes a similar encoder-deconder structure as multi-language machine translation to decode an input image to the corresponding domain, and use the decoded image and the real image an inputs to the discriminator of the CycleGAN. The encoder-decoder structure restricts the number of generator to scale linearly w.r.t. the number of domains, thus is efficient compared to naively applying CycleGAN to a multi-domain setting.

The idea is clear, but from results of the proposed model and of CycleGAN in Figure 2, the proposed model seems to generate images a little more blurrier than CycleGAN. I was expecting to see some quantitative results such as the inception score (though not a good measurement).

---

### Decision · Program_Chairs · 2018-03-20
**ICLR 2018 Workshop Acceptance Decision**

**Decision:**

Accept

**Comment:**

Congratulations, your paper was accepted to the ICLR workshop.